# Exploring sequence landscape of biosynthetic gene clusters with protein language models

**Tatiana Malygina** [1]    **Olga V. Kalinina** [1 2 3]

## Abstract

Many organisms, such as bacteria, fungi, and plants, produce intricate chemicals that are not needed for their growth and reproduction, and thus are called secondary metabolites or natural products (NPs). NPs are a rich source of drugs, with most antibiotics being derivatives of NPs. In a producer organism, NPs are synthesized by a set of enzymes encoded by genes that often lie near each other on the chromosome and are called a biosynthetic gene cluster (BGC).

In this work, we explore the capability of protein language models (PLMs) to produce meaningful representations of BGCs. We employ transfer learning to train models to predict the chemical class of the produced compound and explore the topological properties of the produced embeddings.

The code is available at project's GitHub repository:
https://github.com/kalininalab/NaturalPPLuM.

## 1. Introduction

### 1.1. Natural product discovery with genome mining

Secondary metabolites, or natural products, are produced by many organisms: bacteria, archaea, fungi, plants, and others. Besides their natural role in their producers' corresponding ecosystems, many natural products have been repurposed as medicines (Atanasov et al., 2021). Natural products-derived drugs include most prominently the majority of antibiotics, but also antifungals, cytotoxins, an-

tiprotozoals (Santos-Aberturas & Vior, 2022). In light of the growing menace of microorganisms developing resistance against all available drugs, it is of greatest importance to constantly develop novel compounds with antimicrobial activity (Miethke et al., 2021), and natural product research plays here a major role.

In genomes of secondary metabolite-producing organisms, in particular, bacteria, genes responsible for the synthesis of natural products are located next to each other on the chromosome and expressed as one or a few operons. Such genomic arrangements are called *biosynthetic gene clusters* (*BGCs*). Some BGCs (e.g., non-ribosomal peptide and polyketide synthases) have a highly regular structure consisting of modules, each of which is responsible for adding one unit, supplemented by additional modifying or tailoring enzymes (Schaub et al., 2019). Others, such as ribosomally synthesized and post-translationally modified peptide (RiPP) BGCs, often have a unique composition of modifying enzymes that collectively act to process a precursor peptide into the final product (Kloosterman et al., 2021). Some BGC classes (e.g. alkaloids, terpenes) are defined by specific chemical features of the resulting product, which can be synthesized using various enzymatic pathways; hence corresponding BGCs are heterogeneous on the genetic level. Finally, there are BGCs that combine modules from several classes.

### 1.2. Computational tools for genome mining

The characteristic protein domains required for NPs' biosynthesis and their order in BGCs allow for computational identification of BGCs in genomic sequences. A collection of hidden Markov models describing these domains can be used to detect classical classes of BGCs and hybrid BGCs. This idea is implemented in the state-of-the-art methods antiSMASH (Blin et al., 2023) and PRISM (Skinnider et al., 2020). A caveat of these approaches is that they are only able to detect known types of BGCs. ClusterFinder (Cimermancic et al., 2014) was the first attempt to generalize this idea to unseen combinations of BGC-relevant protein domains, but was not successful in the community.

Recently, machine-learning (ML) and deep-learning (DL) methods are gaining more attention in natural-product re-

[1]Helmholtz Institute for Pharmaceutical Research Saarland (HIPS), Helmholtz Centre for Infection Research (HZI), Saarbruecken, Germany [2]Center for Bioinformatics, Saarland University, Saarbruecken, Germany [3]Faculty of Medicine, Saarland University, Homburg, Germany. Correspondence to: Tatiana Malygina <tatiana.malygina@helmholtz-hips.de>.

*Accepted at the 1st Machine Learning for Life and Material Sciences Workshop at ICML 2024*. Copyright 2024 by the author(s).

search (Mullowney et al., 2023). In particular, for the detection of BGCs, a host of methods became available recently: DeepBGC (Hannigan et al., 2019), GECCO (Carroll et al., 2021), SanntiS (Fragoso et al., 2023). They all employ different DL techniques – bidirectional long short-term memory residual neural networks (DeepBGC), conditional random fields (GECCO), or a combination of bidirectional long short-term memory and convolutional layers (SanntiS) – to predict BGCs from genome sequences. In all cases, they featurize these sequences by predicting genes in them, and then either working directly with their sequence, using transformation such as word2vec (DeepBGC), or predicting domains (Pfam or InterPro) and using these as features.

### 1.3. Protein language models

*Protein language models* (*PLMs*) have been recently developed to create embeddings of protein sequences that can be used in a variety of downstream tasks. Taking inspiration from large language models, PLM is trained to predict the most probable amino acid given the sequence context. Specifically, a fraction of amino acids in a protein's sequence are masked, and the model is trained to predict the identity of masked amino acids. PLMs are typically trained on very large sets of unlabeled sequences, such as around 250 million protein sequences in UniProt (ESM2 (Rives et al., 2019; Lin et al., 2022), Ankh (Elnaggar et al., 2023), etc.)

The major contribution of this study is employing and fine-tuning PLMs for natural product research and exploring the corresponding embedding spaces. In particular, we employ a specific training strategy, termed „leave-one-class-out" or „LOCO" in this work, where all BGCs corresponding to one biosynthetic class are held out from training. We demonstrate that in many cases the trained model can still recognize these BGCs as close entities in the embedding space. This potentially opens the possibility of discovering completely new classes of BGCs.

## 2. Methods

### 2.1. Dataset

As a source of data, we used MIBiG (Terlouw et al., 2022), a publicly available database for experimentally validated BGCs. The latest available version 3.1 has information on 2,502 BGCs associated with one or several biosynthetic classes (Figure 1). There is a class imbalance in the MIBiG dataset; seven major biosynthetic classes dominate the annotations, the most common class being Polyketide, the least common Alkaloid.

We removed all the BGCs with more than one associated biosynthetic class from the dataset, since these samples might create an ambiguity. We selected BGCs which were

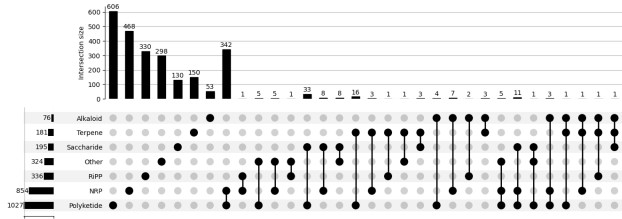

*Figure 1.* Biosynthetic classes in MIBiG 3.1. The UpSet plot indicates what biosynthetic classes are present in the dataset. The bar plot on the left shows number of samples annotated with each of the available biosynthetic classes, with ambiguity: if BGC is annotated with several biosynthetic classes, it will be counted in each of the corresponding categories. The bar plot at the top shows numbers of BGCs with specific combinations of annotations present on the dataset. Each column in the punchcard plot in the middle indicates what biosynthetic class combination corresponds to the bar of bar plot shown above. The leftmost seven columns correspond to the 2035 BGCs which have only one biosynthetic class annotation.

annotated with only one of the main biosynthetic classes: alkaloid (53 BGCs), terpene (150 BGCs), saccharide (130 BGCs), ribosomally synthesized and post-translationally modified peptides (RiPP, 330 BGCs), non-ribosomal peptide (NRP, 468 BGCs), polyketide (606 BGCs), and other (298 BGCs).

For each of the resulting 2035 BGCs we extracted the corresponding sequences: all translated protein sequences or protein domains. Domains were identified from protein sequences with HMMer [1] search using the Pfam-A database (Mistry et al., 2020), following the established procedure described in the literature (Navarro-Muñoz et al., 2020).

### 2.2. Models and Training

We used contrastive learning technique to train Siamese network (Bromley et al., 1994). Thus, we obtained N-dimensional vector representations, known as embeddings, for each BGC-related amino acid sequence. Since input data is represented as protein sequences, as a backbone of our network we used a pre-trained protein language model, with the alphabet consisting of amino acids.

We built our solution on top of the package sentence-transformers (Reimers & Gurevych, 2019) training the Siamese network with a PLM backbone (ESM2 model (Rives et al., 2019; Lin et al., 2022) with 8M parameters, 5 out of 6 layers were frozen during the fine-tuning) and a cosine similarity loss to learn representations of BGCs.

---

[1] http://hmmer.org/

We trained our model to predict whether two sequences of BGCs belong to the same or different biosynthetic classes. For the training we used information from randomly picked pairs of BGCs. We annotated each pair of amino acid sequences with a positive or negative label, depending on whether the corresponding pair of BGCs belongs to the same (positive label) or different (negative label) biosynthetic classes.

ESM2 has the same limitation which is present in many sequential models and in particular in transformer-based architectures: the number of processed tokens in input sequences is limited, extra-tokens are cropped. For ESM2, since each token in the pre-trained alphabet corresponds to one amino acid, it means that only the first 1024 amino acids of the protein sequence are taken into account (as stated both in ESM2 paper and model cards for all available ESM2 models), the remaining data is trimmed. That is the main reason why we did not concatenate all protein sequences to create one protein sequence per BGC. Instead of this, we operated on fragments: protein or domain sequences. If the sequence belongs to some specific BGC, we assign the corresponding biosynthetic class to this sequence.

We considered two settings: with input data being sequences or one protein domain (as predicted with HMMer using a set of HMMs from Pfam-A (Mistry et al., 2020)) or of two consecutive domains. In both cases we processed the data in similar manner, the following preparation steps were performed for each of them.

First, we used one of the two different dataset split strategies to create the train/validation/test splits: (1) Stratified data split: equal fractions of all biosynthetic classes in the training, validation and test sets; (2) LOCO (leave one class out): training and validation sets include all BGCs from six classes with stratified split, the remaining class is held out for testing.

In both cases, data splitting was performed at BGC level: if the BGC was considered to be part of a particular split, all the corresponding amino acid sequences were considered to be the part of that split.

After preparing the splits, we balanced the data as follows:

1. Group sequences by corresponding BGC's biosynthetic class;

2. From each group randomly choose $N$ pairs of sequences, which we consider to be data samples with a positive label. Total number of positive samples is equal to $7 \cdot N$;

3. For each pair of distinct groups, randomly choose $N$ pairs of sequences from corresponding groups, which we consider to be data samples with a negative label;

4. Since the total number of negative samples will be equal to $42 \cdot N$, the dataset will be imbalanced; to fix this, we randomly pick $7 \cdot N$ samples from previously collected samples with negative labels, combine them with all samples with positive label, thus producing a balanced dataset (both in the sense of equal presence of different biosynthetic classes and in the sense of equal number of positive and negative samples).

This balancing procedure ensures that the training dataset has an equal number of positive and negative labels, also, oversampling is done for rare biosynthetic classes, and undersampling is done for common biosynthetic classes.

We used the above procedure to prepare training dataset with $N = 6400$ and validation dataset with $N = 1000$ (the numbers are arbitrary and selected based on approximate epoch training and validation time).

For each of the experiments we used cosine similarity loss during training. To effectively train with this loss function we set positive label value to 1 (for the pair of most similar inputs cosine similarity equals 1), negative value to -1. We train models for 20 epochs or till convergence using sentence transformers library, with the default parameters (AdamW optimizer with learning rate 0.00002).

## 3. Results

### 3.1. Vanilla PLM and fine-tuning with a stratified split

First, we calculated embeddings of each BGCs with a PLM (ESM-2 (Lin et al., 2022)) without further fine-tuning (,,vanilla" embeddings). For this, each BGC was split into domains, and sequences of either each domain or of a concatenated pair of consecutive domains were fed into the PLM. After that, all embeddings corresponding to a single BGC were averaged, producing an embedding for a whole BGC. Second, we fine-tuned the PLM using a stratified split of the dataset, as described in Methods. We computed distances between embeddings of BGCs from each class to the rest of the BGCs (Table 1).

Vanilla embeddings cannot separate biosynthetic classes in a single-domain setup, but can do so to some extent when pairs of consecutive domains are fed into the PLM, although the class admixture is still large. The fine-tuned model trained with the stratified split separates the biosynthetic classes much better, especially with the pair-domain input. This effect is probably largely due to classes of large BGCs, such as NRPs and polyketides, but some other classes with a characteristic domain composition, e.g. RiPPs, can also be well separated (Figures 2, 3).

A closer inspection of distance distributions reveals that fine-tuning with the stratified split leads to a better separation of

*Table 1.* Distances between different BGC classes. Positive silhouette distances indicate that a class is well separated from the other classes in the corresponding embedding space.

| | SILHOUETTE DISTANCE, PAIRS OF DOMAINS | SILHOUETTE DISTANCE, SINGLE DOMAINS | AVERAGE DISTANCE, BIG-SCAPE |
|---|---|---|---|
| WITHOUT FINE-TUNING | 0.098 | -0.1364 | NA |
| STRATIFIED SPLIT | 0.1723 | 0.0048 | NA |
| LOCO SPLITS | | | |
| ALKALOID | -0.0117 | 0.1249 | 0.0022 |
| NRP | 0.2272 | 0.1385 | 0.0840 |
| POLYKETIDE | 0.219 | 0.1726 | 0.0698 |
| RIPP | -0.1649 | -0.1573 | 0.0288 |
| SACCHARIDE | 0.0634 | 0.0772 | 0.0352 |
| TERPENE | -0.1899 | -0.1134 | 0.0220 |
| OTHER | -0.0995 | -0.0943 | -0.0133 |

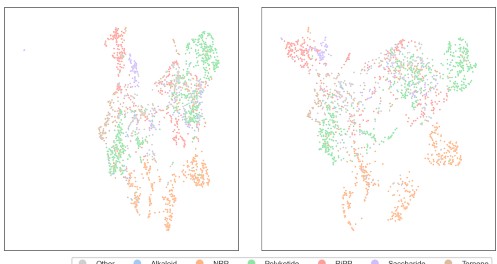

*Figure 2.* Projection of the embeddings spaces with UMAP for vanilla PLM. Single-domain input on the left and pair-domain input on the right. Each dot represents a BGC, colored by biosynthetic class.

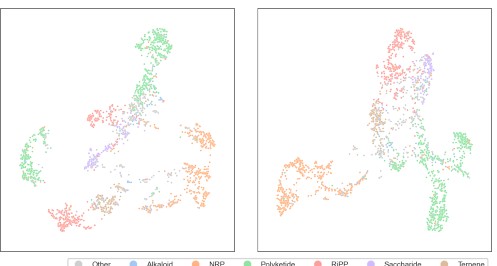

*Figure 3.* Projection of the embeddings spaces with UMAP for the fine-tuned model with a stratified split. Single-domain input on the left and pair-domain input on the right. Each dot represents a BGC, colored by biosynthetic class.

the classes in all cases except terpenes. Despite the fact that NRP and polyketide BGCs have homologous PCP and ACP domains (Pfam family PF00550), they are well separated from each other. RiPP and saccharide BGCs cannot be very well separated, which may be explained by the shared ABC transporter domain (Pfam family PF00005).

### 3.2. Leave-one-class-out (LOCO) models

Further, we considered leave-one-class-out (LOCO) models, in which we left one biosynthetic class completely out of training and used it as a test set. Again, we computed distances between embeddings of BGCs from each class to the rest of BGCs (Table 1). As a baseline, we used distances calculated with the state-of-the-art tool BiG-SCAPE (Navarro-Muñoz et al., 2020). First, despite their wide adoption, BiG-SCAPE distances barely separate different biosynthetic classes from each other. This may indicate that BiG-SCAPE should be applied only to compare BGCs within the same biosynthetic class. Indeed, the weighting coefficients of the three terms of the BiG-SCAPE distance are class-specific (Navarro-Muñoz et al., 2020).

For the LOCO models, single-domain and pair-domain embeddings behave differently for different biosynthetic

classes (Table 1). For large BGCs with a pronounced order of domains, such as NRP and polyketide BGCs, models can group such BGCs together even when they have not seen BGCs of these classes in training, with pair-domain input providing an additional boost (Figure 4B, C). Interestingly, despite a very noticeable presence of homologous domains between the two classes (PCP or ACP domain has to be present in every module of an NRP or polyketide synthase, respectively), the model is able to separate these two classes from each other (Figure 5), probably due to a strong signal that the fine-tuned model detects in these large and regularly built BGCs even without seeing them during training. This is a remarkable observation that indicates a possibility to detect any prominent genetic arrangement that is long enough and contains many repetitive domains.

In all other cases, the pair-domain input does not provide any advantage with respect to separating the unseen class, with only one exception: alkaloids. For this small class with on average few domains per BGC (see below), the LOCO model with the single-domain input achieves a good separation of the unseen BGCs from the rest (Figure 6).

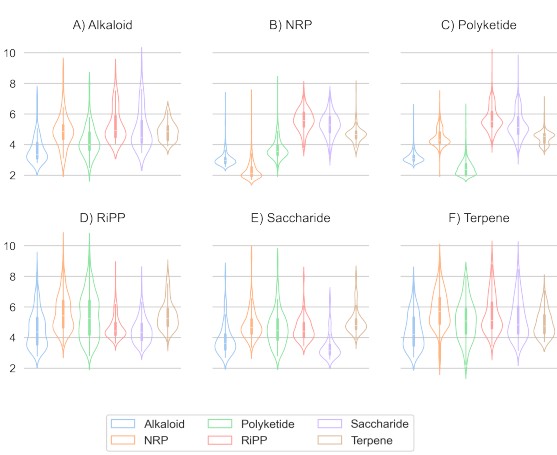

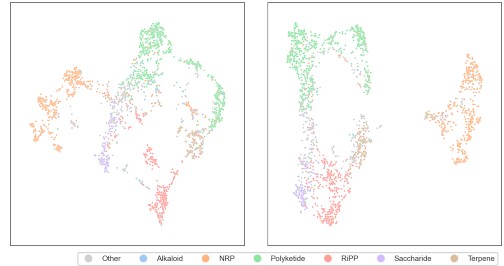

*Figure 6.* UMAP projections of embeddings from the LOCO models with held-out alkaloid BGCs with a single-domain (left) and pair-domain (right) input.

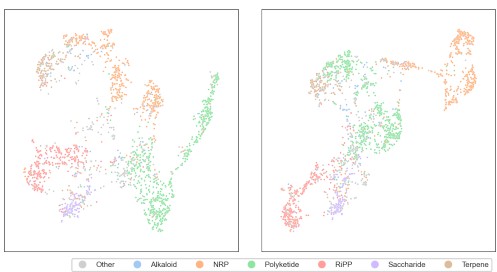

*Figure 4.* Average Euclidean distance from embeddings of BGCs from a specific biosynthetic class X to all other classes from models fine-tuned with LOCO splits, where that particular class X was held out for the test set. Only distances for models with a pair-domain input are shown.

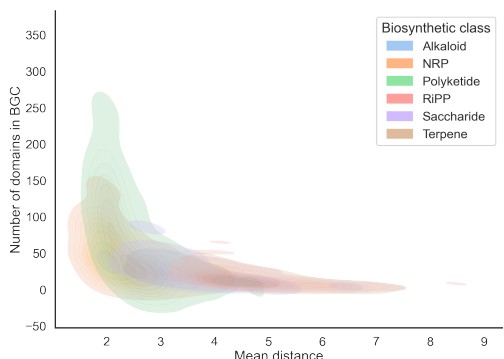

*Figure 7.* Relationship between the number of domains per BGC (vertical axis) and the average distance to other BGCs (horizontal axis) from the same class for embeddings generated by LOCO models for each biosynthetic class.

*Figure 5.* UMAP projections of embeddings from the LOCO models with held-out NRP (left) and polyketide (right) BGCs.

### 3.3. Factors contributing to clustering of embeddings for BGCs from the same biosynthetic class

We set out to investigate what factors contribute to the fact that even in well-separated classes for LOCO models, some BGCs do not cluster with the rest of the class (this corresponds to the upper tails of the distributions in Figure 4). We calculated the number of domains for each BGC and observed a clear inverse correlation between the number of domains per BGC and the average distance to other BGCs of the same class in each of the LOCO models (Figure 7). We do not observe any correlation to the frequency of domains within a biosynthetic class (data not shown).

Finally, we explore the influence of domains that are overrepresented in a specific biosynthetic class. To this end, we calculate the overrepresentation $Over_C^d$ of a Pfam domain $d$ in a class $C$ as $Over_C^d = \frac{N_C^d/N_C}{\sum_C N_C^d/N_C}$, where $N_C^d$ is the number of occurrences of the domain $d$ in class $C$, $N_C$ is the total number of domains in all BGCs from class $C$,

and the sum in the denominator goes over all biosynthetic classes. We weight the number of occurrences by the total number of domain in a certain class to avoid unfairly punishing biosynthetic classes with a small number of BGCs in MiBIG. While doing so, we discarded all domain whose relative occurrence was less than 1%.

Most domains have $Over_C^d$ close to 1 (Figure 8), indicating that they are evenly distributed across different biosynthetic classes. However, some classes have a higher number of overrepresented domains, displaying a skew in the domain composition. For example, the NRP class have highly overrepresented Adenylation domains (PF00501, $Over_{NRP}^{PF00501} = 5.37$) in agreement with the structure of NRP synthetases.

Saccharide and alkaloid biosynthetic classes have the highest number of overrepresented domains. This allows PLMs to detect their own specific domain signature for these two classes, even when no BGCs from them are present in training (Table 1). For alkaloids, the order of domains does not seem to matter, in contrast to NRP and polyketide BGCs, which is evident from the better performance of the single-

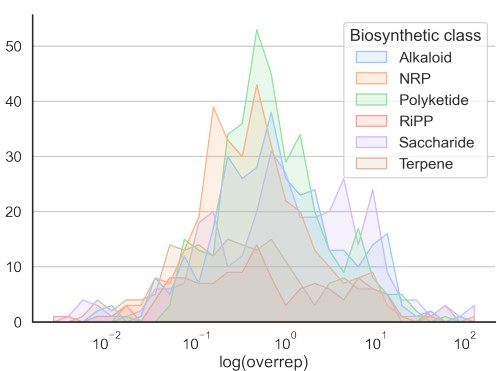

*Figure 8.* Overrepresentation of domains in biosynthetic gene clusters (smoothed histogram).

domain input model. This may be explained by the fact that alkaloids are characteristic plant natural products, whereas the other classes are represented in bacteria, meaning a significant difference in their genetic composition between alkaloid and other BGCs.

## 4. Discussion

In this study, we present an exploratory analysis of PLM-based representation of genetic fragments encoding for the biosynthesis of natural products.

The experiments show that PLM-based embeddings are capable of separating different classes of BGCs corresponding to chemically different classes of compounds. The fine-tuning of one of the protein foundational models allowed us to use all available information on protein universe and to apply it to our specific dataset at the same time. We observed that this combination improved separation between different biosynthetic classes.

Our results of the experiments with the leave-one-class-out (LOCO) cross-validation demonstrated that biosynthetic classes with a well-defined genomic architecture, such as NRP and polyketide synthases, can be detected by the models as a cluster in the embedding space, even when they are not present in training. This means that the model is capable of finding a strong genetic signal, even if it has not observed it before. For the example of NRP and polyketide synthases, despite sharing a homologous domain in their core structure, each class forms a defined cluster in the embedding space even when absent from training. Should another example of a biosynthetic class exist that also has a comparably strong genomic structure, our model should generate embeddings that cluster equally well, and indicates that new unknown BGC classes may potentially be identified with this method.

## 5. Acknowledgements

We are grateful to Amay Ajaykumar Agrawal and Guangyi Chen for inspiring discussion and a critical reading of the manuscript.

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
