# OpenReview forum: "Exploring sequence landscape of biosynthetic gene clusters with protein language models"
_ICML.cc/2024/Workshop/ML4LMS — ML4LMS Poster_

### Official Review · Reviewer_h5S4 · 2024-06-09
**The paper explores protein language models (ESM2) for learning representations of biosynthetic gene clusters.**

**Rating:** 6
**Confidence:** 3

**Review:**

Overall the submission is an interesting study and I’ve a few questions / comments:
- For the siamese network training in section 2.2, I’m wondering if the weights are updated for the language model for only the last layer (keeping the remaining layers fixed) or through the entire model?
- On a related note, do the authors add additional parameters in addition to the PLM?
- Minor: what is silhouette distance in table 1?
- In section 1.3, authors mention “The major contribution of this study is employing embedding created with PLMs to solve the specific task in natural product discovery: predicting the chemical class of a NP from the genomic sequence of thee corresponding BGC.” it seems to me that the results section doesn’t have any direct findings on  prediction  of chemical class? The results are mostly centered around analyzing the embeddings and distances between different classes.
- In the final section, the authors mention that “ new unknown BGC classes may potentially be identified with this method”, I didn’t follow this. I wonder if the authors could provide more details about how would this be possible assuming we have a trained model with the setting as described.

---

### Official Review · Reviewer_HHop · 2024-06-11
**an LLM to identify new biosynthetic gene clusters**

**Rating:** 7
**Confidence:** 3

**Review:**

1. Quality: high-quality work to build and validate the LLM
2. Clarity: well-written text and figures
3. Originality: this appears to be a novel application
4. Significance of this work: Identification of enzymes and synthesis route for novel chemistry is very significant for healthcare and potentially green chemistry. this work could lead to the identification of novel pathways.
5. Pros:
6. Cons:No sure clear metric to determine the success of this work

---

### Official Review · Reviewer_QpP3 · 2024-06-12
**Review of xploring sequence landscape of biosynthetic ...**

**Rating:** 4
**Confidence:** 4

**Review:**

Might be an interesting biological application; however, the technical novelty is limited.

"For ESM2, since each token in the pre-trained alphabet corresponds to one amino acid, it means that only the first 512 amino acids of the protein sequence are taken into account, and the remaining data is trimmed." I do not understand where this is coming from. According to the ESM2 paper, the context is 1024 residues long.